# PRISM: PRogressive dependency maxImization for Scale-invariant image Matching

## ABSTRACT

Image matching aims at identifying corresponding points between a pair of images. Currently, detector-free methods have shown impressive performance in challenging scenarios, thanks to their capability of generating dense matches and global receptive field. However, performing feature interaction and proposing matches across the entire image is unnecessary, as not all image regions contribute beneficially to the matching process. Interacting and matching in unmatchable areas can introduce errors, reducing matching accuracy and efficiency. Furthermore, the scale discrepancy issue still troubles existing methods. To address above issues, we propose PRogressive dependency maxImization for Scale-invariant image Matching (PRISM), which jointly prunes irrelevant patch features and tackles the scale discrepancy. To do this, we first present a Multi-scale Pruning Module (MPM) to adaptively prune irrelevant features by maximizing the dependency between the two feature sets. Moreover, we design the Scale-Aware Dynamic Pruning Attention (SADPA) to aggregate information from different scales via a hierarchical design. Our method's superior matching performance and generalization capability are confirmed by leading accuracy across various evaluation benchmarks and downstream tasks. The code will be publicly available.

## CCS CONCEPTS

• **Computing methodologies** → **Vision for robotics**; **Matching**; **Scene understanding**.

## KEYWORDS

Image Matching, Patch Pruning, Scale-Aware

## 1 INTRODUCTION

Image matching, a pivotal task in computer vision, finds extensive applications in areas such as image stitching [30, 41], visual localization [49, 51], 3D reconstruction [54, 55], etc. Traditionally, it involves identifying corresponding points across two images, which is a detector-based paradigm [3, 34, 48]. This paradigm follows a three-step pipeline: (1) keypoint detection, (2) keypoint description, and (3) matching based on descriptor similarity. Although detector-based methods are generally effective and efficient, they falter in complex environments such as poor texture, repetitive patterns and

Unpublished working draft. Not for distribution.

Permission to make digital or hard copies of all or part of this work for personal or classroom use is granted without fee provided that copies are not made or distributed for profit or commercial advantage and that copies bear this notice and the full citation on the first page. Copyrights for components of this work owned by others than the author(s) must be honored. Abstracting with credit is permitted. To copy otherwise, or republish, to post on servers or to redistribute to lists, requires prior specific permission and/or a fee. Request permissions from permissions@acm.org.

*ACM MM, 2024, Melbourne, Australia*

© 2024 Copyright held by the owner/author(s). Publication rights licensed to ACM.
ACM ISBN 978-x-xxxx-xxxx-x/YY/MM
https://doi.org/10.1145/nnnnnnn.nnnnnnn

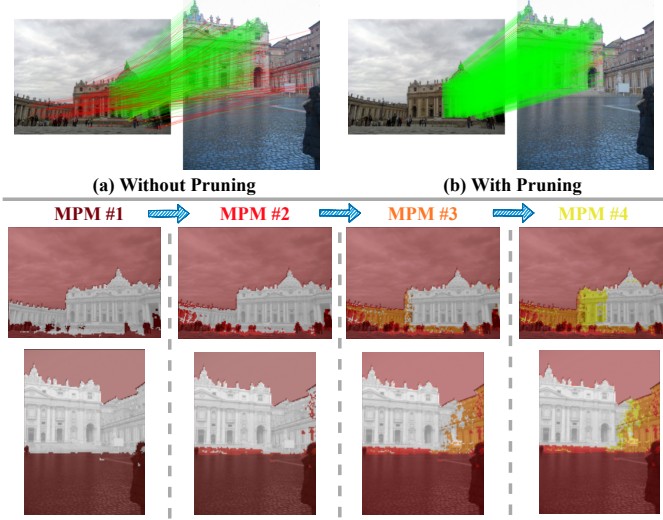

**(a) Without Pruning**      **(b) With Pruning**

MPM #1 → MPM #2 → MPM #3 → MPM #4

**(c) The evolution of pruning masks**

**Figure 1: The basic idea of our proposed methods.** Given two images, not all image patches are helpful to the matching process. Conducting feature interactions and searching matches across the entire image can be detrimental (Without Pruning). We propose to gradually prune the irrelevant patches by maximizing the dependency between two images, resulting in more robust and accurate matches (With Pruning). (c) shows the pruning masks estimated by the corresponding MPM. MPM adaptively prunes irrelevant patches from shallow to deep layers ■ → ■ → ■ → ■. Feature interactions and match searches are only conducted in the white mask regions.

significant viewpoint changes, where keypoint detection may not yield sufficient keypoints.

To address the limitation, researchers propose deep learning-based methods to improve the reliability of traditional pipeline. Superpoint [13], Lift [71] and some other works [14, 36, 44] enhance the repeatability and distinguishability of the keypoints. SuperGlue [52] and its variations [32, 56] propose transformer-based [66] matchers, jointly matching sparse points and rejecting outliers. Although these works gain improvements in more reliably matching, they still share the limitations of the detector-based paradigm. The keypoint detection is generally a bottleneck and the performances in some complex environments (e.g., poor texture, repetitive patterns and significant viewpoint changes) are unsatisfactory.

To address these issues, another line of works [28, 45, 46] abandons the keypoint detection, so-called detector-free methods. They first use CNN [31] to extract image features. In order to generate dense matches, these approaches exhaustively search for matches across entire feature maps. Potential patch-level matches are proposed from the whole feature map by constructing correlation cost

volume between two feature maps. The coarse matches are then filtered by threshold and Mutual Nearest Neighborhood criterion. Finally, the valid coarse matches are further refined into pixel-level matches. Considering the feature maps generated by CNN have a limited receptive field, LoFTR [59] and its variants [10, 18, 72] utilizes transformer backbone [24] to model long-range dependencies, resulting in better feature matching performance.

While detector-free methods address the limitations of detector-based methods, these techniques still encounter significant challenges. On the one hand, detector-free methods suffer from irrelevant feature interference. Feature interactions and match searches may perform in unnecessary areas, leading to erroneous matches, as shown in Figure 1. Notably, for a pair of images divided into $M$ and $N$ patches, this matching paradigm proposes $M \times N$ potential matches. However, the maximum viable matches are only $max(M, N)$ in theory. This discrepancy suggests that a significant proportion of these matches are erroneous (such as matches in unmatchable areas like sky and clouds or non-overlapping regions). TopicFM [18] and AdaMatcher [22] utilize semantic and co-visible information to filter match proposals. But the feature interactions still span the entire image.

On the another hand, detector-free methods fall short in addressing the scale discrepancy. As detector-free methods generate matches in pixel-level, exhaustively searching matches across different scales can bring unbearable computational costs. Although ASTR [72] and AdaMatcher [22] attempt to estimate the scale variation between images via patch-level matching and adjust the patch size in fine matching process based on the estimated scale ratio, the patch-level matching itself can also be erroneous due to the scale problem. PATS [40] introduces an innovative yet time-intensive framework that iteratively rescales and segments images into smaller, scale-consistent patches to mitigate the scale issue, but the time consumption is significantly increased.

To deal with the above challenges, we propose a novel framework, PRogressive dependency maxImization for Scale-invariant image Matching (PRISM), which simultaneously prunes irrelevant patches and tackles scale difference. The basic idea is illustrated in Figure 1. The key innovation is to prune unnecessary image patches adaptively and gradually, and model the scene of various scales simultaneously within the same attention mechanism. To eliminate the interference of the irrelevant features, we propose the Multi-scale Pruning Module (MPM) to dynamically prune irrelevant features by gradually maximizing the dependency between the two feature sets, where the dependency is usually measured by Mutual Information. By pruning irrelevant features gradually, the computational resources can be focused on those features that are most informative and relevant for matching. In addition, to solve the scale discrepancy, we propose the novel Scale-Aware Dynamic Pruning Attention (SADPA) mechanism, which injects the scale space analysis into the attention mechanism via a hierarchical design and focuses attention on the selected features. This scheme gives SADPA favorable computational efficiency alongside the ability to model multi-scale features.

Experimental evaluations show that PRISM sets a new state-of-the-art (SOTA), surpassing both detector-based and detector-free baselines across various tasks, such as homography estimation, relative pose estimation, and visual localization. Experiments also showcase PRISM's robust generalization capabilities. Our ablation studies verify the effectiveness of the proposed MPM and SADPA. The key contributions are summarized as follows:

- A novel image matching framework PRISM is proposed, employing a Multi-scale Pruning Module (MPM) to aggregate information in different scales and prune irrelevant features by maximizing the dependency gradually.
- A Scale-Aware Dynamic Pruning Attention (SADPA) is proposed, which dynamically adjusts the attention focus and aggregates information across multiple scales.
- PRISM is demonstrated to achieve state-of-the-art results across a comprehensive set of benchmarks, showcasing its robust generalization capabilities across diverse datasets.

## 2 RELATED WORK

### 2.1 Detector-based image matching

Detector-based image matching has been studied for several decades. It involves identifying keypoints in images and finding their correspondences across different views. The evolution of detector-based image matching began with foundational techniques like the Harris Corner Detector [20], advancing to sophisticated systems such as SIFT [34] for improved scale and rotation resilience. Developments in detection speed and reliability followed [3, 47, 48]. With the success of Convolutional Neural Networks (CNN) in the field of image processing [21, 57, 60], numerous researchers have begun to employ CNNs for keypoint detection and description, achieving impressive results. Superpoint [13] jointly trains the detector and descriptor, largely improving the accuracy and robustness of matching. Many works follow this line, further improving the reliability and uniqueness of the keypoints [14, 27, 35, 36, 44, 65, 69].

After detecting and describing keypoints, matching typically involves a nearest neighbor search in descriptor space, complemented by heuristic filters like Lowe's ratio test [34]. However, this approach struggles under challenging conditions. SuperGlue [52] addresses this by employing an Attentional Graph Neural Network to jointly match features and filter outliers. However, it has quadratic complexity with the number of keypoints due to the attention mechanism. Subsequent efforts, such as SGMNet [9], ClusterGNN [56], and LightGlue [32], have aimed to reduce computational load through strategies like initializing with a subset of reliable matches, dividing keypoints into subgraphs, and adapting network depth and width based on image pair difficulty. However, the dependency on keypoint and descriptor repeatability limits the robustness of detector-based methods against extreme variations in viewpoint, repetitive patterns, and texture-less surfaces.

### 2.2 Detector-free image matching

Detector-free image matching methods eschew the keypoint detection step and directly generate dense matches from the image. Early works are cost volume-based, such as NCNet series [28, 45, 46]. They use CNN to extract dense feature maps and construct 4-D cost volume to exhaust all potential matches. Although they have made some progress, the receptive field of CNN is limited, and the image resolution is restricted due to the expensive computational cost. Recently, LoFTR [59] extends the limited receptive field to global consensus with the help of the global receptive field and long-range

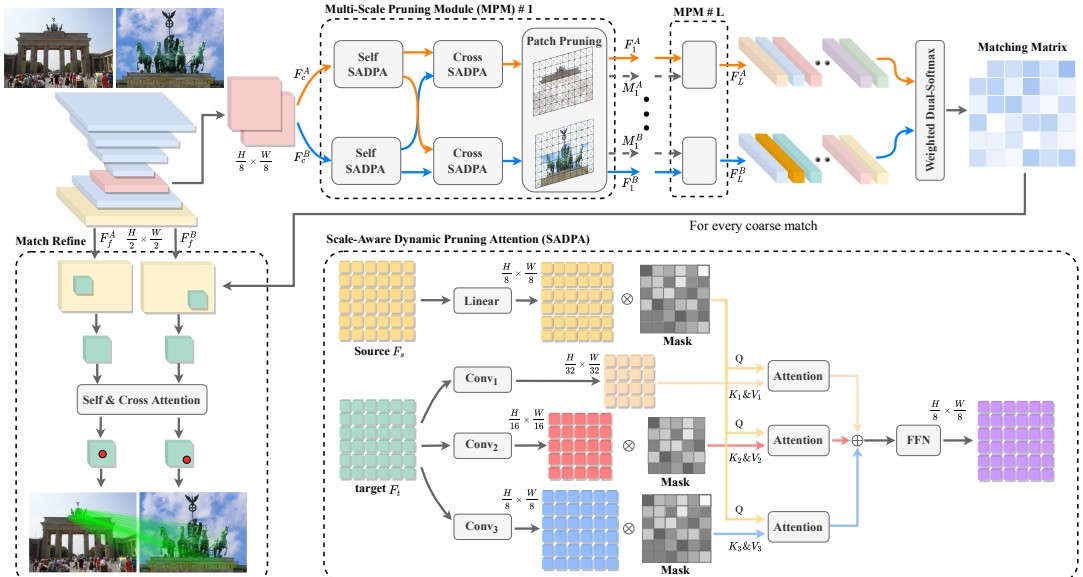

**Figure 2: Overview of PRISM.** PRISM starts from a CNN-based backbone to extract coarse-level $F_c^A, F_c^B$ and fine-level features $F_f^A, F_f^B$. $F_c^A, F_c^B$ are fed into the proposed iterative Multi-scale Pruning Module (MPM) for updating and pruning (Sec. 3.2). In each MPM layer, the features are first transformed by the self- and cross- SADPA with a hierarchical design (Sec. 3.2.2) to aggregate information from selected features of various scales. Then the Patch Pruning module (Sec. 3.2.1) eliminates irrelevant features to maximize the NMI between the two feature sets. After $L$ MPM blocks, the final features $F_L^A$ and $F_L^B$ are used to acquire the coarse matching Matrix by Weighted Dual-softmax (Sec. 3.3). Finally, we use the mutual nearest neighbor strategy and the threshold $\theta_c$ to filter the valid coarse matches $\mathcal{M}_c$. Then $\mathcal{M}_c$ are projected to fine level features maps $F_f^A, F_f^B$ and refined to sub-pixel precision matches $\mathcal{M}_f$.

dependencies of Transformers [66]. However, LoFTR and its successors [7, 16, 70, 76] still encounter significant challenges, particularly regarding the scale disparity problem and the distraction issue of linear attention. AdaMatcher [22] and ASTR [72] estimate the scale variation via coarse matching results and resize the patch size by scale ratio, but they ignore that the coarse level matching itself can be erroneous due to the scale problem. PATS [40] models the scale problem as a patch area transportation problem and designs an iterative framework to find matches from coarse to fine, but the time cost is unacceptable. ASpanFormer [10] uses estimated optical flow to guide the attention span. ASTR [72] proposes a spot-guided attention framework to restrict the feature aggregations. Quadtree attention [63] builds token pyramids and computes attention in a coarse-to-fine manner. However, they neglect the comprehensive understanding of scene semantics and contextual relationships, achieving limited enhancements. TopicFM [18] constrains matches to regions with identical semantics, but it does not account for completely unmatchable categories(e.g., sky, clouds) or non-overlapping regions. Other works [23, 62] that try to directly decode the coordinates of matching keypoints are also related.

### 2.3 Mutual Information based Feature Selection

Feature selection involves choosing a subset of the available features based on specific criteria to eliminate irrelevant, redundant, or noisy features, which is a crucial aspect of data mining. The use of Mutual Information (MI) to assess the dependency among features for the purpose of feature selection was first introduced in [2], referred to as Mutual Information Maximization. MI assesses the information contribution of variables towards the learning goal. Several methods [5, 37] leveraging MI for feature selection have been proposed to enhance performance across diverse learning tasks. One notable approach is the minimal-redundancy-maximal-relevance (mRMR) criterion [43], which uses average MI as a criterion and selects features with a trade-off between dependency and redundancy of the selected features. Further advancing the field, the Normalized Mutual Information Feature Selection (NMIFS) technique was introduced to mitigate MI's bias towards multi-valued features by normalizing MI (NMI) values to a [0,1] range [15]. This method uses average NMI to select features, improving upon the standard MI approach by addressing its inherent bias. Subsequently, several methods [17, 19, 68, 68] have been proposed to further enhance the performance.

MI distinguishes itself from other dependency measures by its ability to quantify any relationship between variables and its stability under space transformations like rotations and translations [15, 25]. This makes MI-based feature selection particularly appealing for patch feature pruning in the context of image matching. It underpins our study's innovative patch pruning technique, aimed at eliminating irrelevant patch features.

## 3 METHOD

### 3.1 Overview

Given a pair of images, our task is to identify reliable matches across images. To address the scale discrepancy and reduce the interference

of irrelevant features, we propose to gradually eliminate redundant image patches through adaptive pruning and simultaneously model scenes of various scales within the same attention framework. The overview of PRISM is shown in Figure 2. Mathematically, for image $\mathbf{I}^A$ and $\mathbf{I}^B$, PRISM generates matches as in Algorithm 1:

---

**Algorithm 1** PRISM: PRogressive dependency maxImization for Scale-invariant image Matching

---

1: **Input:** a pair of images $\mathbf{I}^A$ and $\mathbf{I}^B$
2: **Output:** Matches $\mathcal{M}_f$
3: $F_c^A, F_c^B, F_f^A, F_f^B = \text{CNN}(\mathbf{I}^A, \mathbf{I}^B)$
4: $M_0^A, M_0^B \leftarrow$ Masks of all 1 with the same size of $F_c^A, F_c^B$
5: $F_0^A, F_0^B = F_c^A, F_c^B$
6: **for** $l = 1$ to $L$ **do**
7: $\quad F_l^A, F_l^B = f_{\text{transform}}(F_{l-1}^A, F_{l-1}^B, M_{l-1}^A, M_{l-1}^B)$
8: $\quad M_l^A = \underset{M_l^A}{\arg\max}\, \text{D}(\hat{F}_l^A, F_l^B), M_l^B = \underset{M_l^B}{\arg\max}\, \text{D}(\hat{F}_l^B, F_l^A)$
9: **end for**
10: $\mathcal{M}_c = f_{\text{matching}}(F_L^A, F_L^B)$
11: $\mathcal{M}_f = f_{\text{refine}}(\mathcal{M}_c, F_f^A, F_f^B)$
12: **return** $\mathcal{M}_f$

---

In PRISM, each **for** loop constitutes a Multi-scale Pruning Module (MPM), and there are total $L$ MPMs. Each MPM takes the last MPM's output, i.e., $F_{l-1}^A, F_{l-1}^B, M_{l-1}^A, M_{l-1}^B$ as input and conducts $f_{\text{transform}}$ and dependency maximization. $M_l^A$ is the mask to select the best subset $\hat{F}_l^A$: $\hat{F}_l^A = M_l^A \otimes F_l^A$, that maximizes the dependency $D(\cdot)$ between the two feature sets. $M_l^B$ is defined in a similar way.

## 3.2 Multi-scale Pruning Module

Given the coarse-level features maps $F_c^A$ and $F_c^B$ at $\frac{1}{8}$ resolution, the Multi-scale Pruning Module extracts multi-scale features and progressively eliminates irrelevant features. Specifically, in each MPM, the coarse feature maps are first transformed by $f_{\text{transform}}(\cdot)$, which consists of a self SADPA and a cross SADPA. Then, the Patch Pruning module eliminates irrelevant features by maximizing the dependency between the two feature sets, resulting in two pruning masks for use in the next layer. We first introduce the Patch Pruning part in MPM.

*3.2.1 **Patch Pruning**.* As stated in Introduction (Sec. 1), existing methods [7, 10, 22, 59, 72] may search matches and perform feature interactions in unmatchable areas. It harms the matching accuracy and increases the computational cost. A pruning module is needed to exclude irrelevant patch features.

Inspired by the Mutual Information based Feature Selection methods [15, 43], we innovatively propose to identify the most characterizing features by maximizing the dependency between the two feature sets in the context of image matching. Specifically, in $l$th MPM layer, the patch pruning module takes the transformed feature maps $F_l^A$ and $F_l^B$ as input, where $|F_l^A| = M$ and $|F_l^B| = N$. For $F_l^A$, our target is to find a feature subset $\hat{F}_l^A \subseteq F_l^A$ which has the largest dependency on $F_l^B$:

$$M_l^A = \underset{M_l^A}{\arg\max}\, \text{D}(\hat{F}_l^A, F_l^B) \qquad (1)$$

The dependency $D(\cdot)$ is usually characterized in terms of mutual information (MI) as follows:

$$D(\hat{F}_l^A, F_l^B) = I(\{\hat{F}_l^A(i)|i = 1, ..., m\}; F_l^B), \hat{F}_l^A(i) \in \hat{F}_l^A \qquad (2)$$

where $F_l^B$ can be treated as a multivariate variable and $I(\cdot)$ denotes the MI. Given two random variables $X, Y$, the MI is defined as:

$$I(X; Y) = D_{KL}(p(x, y) \| p(x)p(y)) = \iint p(x, y) \log \frac{p(x, y)}{p(x)p(y)} dx dy.$$

In this formula, $D_{KL}$ is the KL-divergence between $p(x, y)$ and $p(x)p(y)$, which represents the joint distribution and product of the marginal distributions of $x$ and $y$, respectively.

However, the Max-Dependency is hard to implement in high-dimensional space, and searching the feature subspaces exhaustively costs $O(2^M)$ [43]. An alternative is to select features based on Max-Relevance, which approximates the Max-Dependency with the mean value of all mutual information values as in [43]:

$$D(\hat{F}_l^A, F_l^B) \approx \frac{1}{|\hat{F}_l^A|} \sum_{\hat{F}_l^A(i) \in \hat{F}_l^A} I\left(\hat{F}_l^A(i); F_l^B\right). \qquad (3)$$

According to this equation, we can increase the Max-Dependency by eliminating patch features with low MI. The MI between a patch feature $\hat{F}_l^A(i)$ and another feature set $F_l^B$ can be expressed compactly in terms of multi-information as in [39, 67]: $I\left(\hat{F}_l^A(i); F_l^B\right) = \sum_{k=1}^{N} \sum_{\substack{\forall S \subseteq F_l^B \\ |S|=k}} I([S \cup \hat{F}_l^A(i)])$ where $I([S \cup \hat{F}_l^A(i)] = I(s_1; s_2; \cdots; s_k; \hat{F}_l^A(i))$. Note that the sum on the right side is taken over all subsets $S$ of size $k$ drawn from the feature set $F_l^B$. To standardize the measure of shared information between variables, we utilize Normalized MI (NMI) as in [15]:

$$\text{NMI}\left(\hat{F}_l^A(i); F_l^B\right) = 2 \frac{I\left(\hat{F}_l^A(i); F_l^B\right)}{H(\hat{F}_l^A(i)) + H(F_l^B)} \in [0, 1] \qquad (4)$$

$\text{NMI}\left(\hat{F}_l^A(i); F_l^B\right)$ quantifies the amount of information that $\hat{F}_l^A(i)$ shares with $F_l^B$. This concept is intrinsically linked to the fundamental principle of matching, where a pair of features is considered to be matched when they describe the same scene, signifying that their information is shared.

However, MI and NMI have historically been difficult to compute [42]. Exact computation is only tractable for discrete variables or a limited family of problems where the probability distributions are known. Inspired by MINE [4], we propose to learn a neural network to estimate the NMI:

$$\text{NMI}\left(\hat{F}_l^A(i); F_l^B\right) \approx \text{Sigmoid}(\Phi_l(f_{\text{transform}}(F_{l-1}^A, F_{l-1}^B, M_{l-1}^A, M_{l-1}^B)(i))) \qquad (5)$$

where $\Phi_l$ represents an MLP at the last of the $l$th layer MPM module. To maximize the dependency, we prune the features whose NMI is lower than a threshold $\theta_p = 0.05$. The mask for the locations of the removed features is set to 0, resulting in the updated pruning mask $M_l^A$. $M_l^B$ is obtained in a similar way.

The Patch Pruning module plays a critical role by effectively reducing the search space for potential matches, enhancing both accuracy and efficiency. The structure and computational flow of these components ensure that as the matching process evolves, the algorithm becomes increasingly focused on the most promising match candidates. Figure. 3 shows the effect of this module. We can see most of the invaluable regions such as the sky with less

**Figure 3: Visualization of pruning masks and the matching results on MegaDepth dataset.** The Patch pruning can identify redundant image patches and exclude them from subsequent feature interactions gradually (from shallow to deep layers ■ → ■ → ■ → ■). It avoids most incorrect matches.

mutual information are pruned. This improves both the precision and computational efficiency of the image matching process.

#### 3.2.2 Scale-Aware Dynamic Pruning Attention.

**self and cross SADPA.** In each MPM, we use a succession of one self SADPA and one cross SADPA to update the features, as shown in Figure 2. The $l$th MPM takes feature maps $F_{l-1}^A, F_{l-1}^B$ and masks $M_{l-1}^A, M_{l-1}^B$ as input, where $F_{l-1}^A, F_{l-1}^B$ are first input to two self-SADPAs respectively. The cross SADPA takes the output of two self SADPAs as input to further interact features across images. For clarity of presentation, we denote the input feature maps of SADPA as $F_s$ and $F_t$ respectively. Thanks to this design, MPM not only enhances the model's ability to capture intra-image information but also broadens the understanding of inter-image relationships.

**Preliminaries.** Before delving into SADPA, let's briefly introduce the commonly used vanilla attention. The vanilla attention takes three input vectors: Query $Q$, Key $K$ and value $V$. The $Q$ queries information from the $K - V$ pairs according to the similarity matrix between $Q$ and $K$: Attention$(Q, K, V) = $ softmax$(QK^T)V$. However, the size of the weight matrix softmax$(QK^T)$ increases quadratically with the image size and the computational cost is unacceptable. As a result, directly applying scale-space analysis by conducting attention between features of different scales is infeasible. The absence of the scale-space analysis impedes the capability of the models to capture multi-scale scenes. Although existing methods [10, 18, 59] optimize the quadratic complexity to linear using linear attention [24], it comes at the cost of sacrificing representational capability [6] and matching accuracy [63, 72]. Therefore, we introduce our Scale-Aware Dynamic Pruning Attention, which injects the scale space analysis into the attention mechanism.

**SADPA.** The design of SADPA is shown in the right-down part of Figure. 2, which performs attention at different scales in parallel. SADPA takes the source and target feature maps $F_s$ and $F_t$ and their

corresponding pruning masks $M_s$ and $M_t$ as input. SADPA uses a Linear module to project $F_s$ into Query and then trims unnecessary source features by the mask $M_s$:

$$Q = \text{Linear}(F_s) \otimes M_s \qquad (6)$$

where $\otimes$ denotes element-wise mask operation and Linear$(\cdot)$ is a learned linear transformation. SADPA captures multi-scale features by downsampling the target feature map $F_t$ to construct a 3-level feature pyramid using convolution layers with varying kernel sizes and strides. Specifically, $F_t$ is reduced to $\frac{1}{32}$ resolution to be the coarsest Key and Value. No pruning mask is applied to model the long-range dependencies and large scenes:

$$K_i = V_i = \text{Conv}_i(F_t), i = 1 \qquad (7)$$

where Conv$_i$ means a convolutional operator with kernel size and stride of $r_i$. The other two layers in the pyramid are downsampled into $\frac{1}{16}$ and $\frac{1}{8}$ resolution separately to encode the local neighborhood constraints and small scenes. In order to reduce the computational cost and avoid disruption caused by irrelevant features, pruning masks is applied to prune irrelevant features. We downsample the pruning masks to the same size as the feature maps:

$$K_i = V_i = \text{Conv}_i(F_t) \otimes \text{down}_i(M_t), i \in \{2, 3\} \qquad (8)$$

where down$_i$ denotes nearest-neighbor interpolation by the ratio $r_i$. Then, the attention is performed at different scales and the retrieved messages $m_i$ from different scales are concatenated and fused with an FFN to update the source features:

$$m_i = \text{Attention}(Q, K_i, V_i), i \in \{1, 2, 3\},$$
$$F_{out} = \text{FFN}(m_1 \oplus m_2 \oplus m_3, F_s). \qquad (9)$$

So SADPA injects the scale space analysis into the attention mechanism by projecting the $K$ and $V$ into different scales via convolutions before computing the attention matrix. It allows attention processing at various scales: fine-level attention retains more local details, whereas coarse-level attention captures broader image contexts. By excluding the irrelevant features during the fine-level attentions, the computational cost and disruption are largely reduced.

**Positional Encoding.** The spatial relationship of features is crucial for matching. But the attention mechanism falls short in recognizing spatial positional relationships. Therefore, a positional encoding is necessary. Previous methods [10, 18, 59] use the 2D extension of the standard sinusoidal encoding following DETR [8]. However, in the context of two-view geometry, it's apparent that the positioning of visual elements changes consistently in relation to camera movements within the image plane. This phenomenon underscores the need for an encoding that prioritizes relative position over absolute position. we adopt the Rotary Position Embedding (RoPE) [58] to remedy this problem. It allows the model to effectively identify the relative positioning of point $j$ from point $i$. We only apply RoPE in self SADPA, because it makes no sense to compute the relative positions across images.

### 3.3 Match Prediction by Weighted Dual-Softmax

After the update by $L$ MPM blocks, we get the final transformed features $F_L^A$ and $F_L^B$ and flatten them for further use. We also obtain their corresponding estimated NMI $\sigma_L^A$ and $\sigma_L^B$ at the last MPM layer. We calculate the matching matrix $\mathcal{P}$ combining both the

similarity and the estimated NMI:

$$\mathcal{P}(i,j) = \sigma_L^A(i)\sigma_L^B(j)\text{softmax}(S(i,\cdot))_j \cdot \text{softmax}(S(\cdot,j))_i \quad (10)$$

where $S$ is the similarity matrix computed by the features: $S(i,j) = \tau \cdot \langle F_L^A(i), F_L^B(j) \rangle$. $\tau$ is the temperature coefficient. The NMI is used to weight the matching matrix, as the valid match points should be both relevant and similar. We selected matches with $\mathcal{P}(i,j) > \theta_c$ and filtered them using the mutual nearest neighbor strategy, resulting the coarse matches $\mathcal{M}_c$.

## 3.4 Supervision

The final loss is composed of three parts: coarse matching loss $\mathcal{L}_c$, sub-pixel refinement loss $\mathcal{L}_f$ and patch pruning loss $\mathcal{L}_p$:

$$\mathcal{L} = \mathcal{L}_c + \mathcal{L}_f + \mathcal{L}_p \quad (11)$$

**Coarse Matching Loss.** We use cross entropy loss to supervise the coarse matching matrix $\mathcal{P}$:

$$\mathcal{L}_c = -\frac{1}{|\mathcal{M}_c^{gt}|} \sum_{(i,j) \in \mathcal{M}_c^{gt}} \log \mathcal{P}(i,j) \quad (12)$$

The ground-truth coarse matches $\mathcal{M}_c^{gt}$ is calculated from the camera poses and depth maps at coarse resolution.

**Sub-pixel Refinement Loss.** Following LoFTR [59], we use the L2-distance between each refined coordinate and the ground truth reprojection coordinate and normalize it by the coordinate variance $\phi$:

$$\mathcal{L}_f = \frac{1}{|\mathcal{M}_f|} \sum_{(i,j') \in \mathcal{M}_f} \frac{1}{\phi^2(\hat{i})} \left\| \hat{j}' - \hat{j}'_{gt} \right\|_2, \quad (13)$$

We compute the the $\hat{j}'_{gt}$ by warping $\hat{i}$ on $\mathbf{I}^A$ to $\mathbf{I}^B$ with the ground-truth pose and depth.

**Patch Pruning Loss.** We supervise the Patch Pruning module as the negative log-likelihood loss over the estimated NMI for all features. Patch features derived from $\mathbf{I}^A$ that can find matches in $\mathbf{I}^B$ are defined as $A_m$, and the rest patch features can not find matches are defined as $A_n$. $B_m$ and $B_n$ is defined similarly. For NMI estimated at $l$-th MPM layer, the loss $\mathcal{L}_p^A(l)$ and $\mathcal{L}_p^B(l)$ is defined as:

$$\mathcal{L}_p^A(l) = -\left(\frac{1}{|A_m|} \sum_{i \in A_m} \log(\sigma_l^A(i)) + \frac{1}{|A_n|} \sum_{i \in A_n} \log(1 - \sigma_l^A(i))\right)$$

$$\mathcal{L}_p^B(l) = -\left(\frac{1}{|B_m|} \sum_{j \in B_m} \log(\sigma_l^B(j)) + \frac{1}{|B_n|} \sum_{j \in B_n} \log(1 - \sigma_l^B(j))\right)$$

$$(14)$$

We supervise it at every MPM layer. The final $\mathcal{L}_p$ is defined as:

$$\mathcal{L}_p = \frac{1}{L} \sum_l \frac{\mathcal{L}_p^A(l) + \mathcal{L}_p^B(l)}{2} \quad (15)$$

## 3.5 Implementation Details

We adopt the same ResNet-FPN [21, 31] architecture as LoFTR [59] to extract image features. The dimension for coarse features and fine features are 256 and 128, respectively. We use 4 MPM layers for feature updating and pruning. For each SADPA, the convolutional kernel sizes are 4, 2 and 1 from coarse to fine. We use an efficient implementation of attention [12] and each attention unit has 4 heads. We only train the model on the MegaDepth [29] dataset without any data augmentation and test it on all datasets and tasks

| Category | Methods | Homography AUC | | |
|---|---|---|---|---|
| | | @3px | @5px | @10px |
| Sparse | SP [13]+SG [52] | 53.9 | 68.3 | 81.7 |
| | D2Net [14]+NN | 23.2 | 35.9 | 53.6 |
| | R2D2 [44]+NN | 50.6 | 63.9 | 76.8 |
| | Patch2Pix [75] | 46.4 | 59.2 | 73.1 |
| | DISK [65]+NN | 52.3 | 64.9 | 78.9 |
| Dense | DRC-Net [28] | 50.6 | 56.2 | 68.3 |
| | Sparse-NCNet [45] | 48.9 | 54.2 | 67,1 |
| | LoFTR [59] | 65.9 | 75.6 | 84.6 |
| | Quadtree [63] | 66.3 | 76.2 | 84.9 |
| | ASpanFormer [10] | 67.4 | 76.9 | 85.6 |
| | 3DG-STFM [38] | 64.7 | 73.1 | 81.0 |
| | ASTR [72] | 71.7 | 80.3 | 88.0 |
| | TopicFM [18] | 70.9 | 80.2 | **88.3** |
| | Efficient LoFTR [70] | 66.5 | 76.4 | 85.5 |
| | PRISM(Ours) | **71.9** | **80.4** | **88.3** |

Table 1: Homography estimation on Hpatches.

to demonstrate the generalization ability. We follow the same train-test split as in LoFTR [59]. We use the AdamW [33] optimizer with an initial learning rate of $8 \times 10^{-4}$. The entire model is trained end-to-end with a batch size of 24 on 8 NVIDIA A100, taking 1.5d to converge.

## 4 EXPERIMENTS

### 4.1 Homography Estimation

Homography is crucial in two-view geometry. It enables the transformation of perspectives between two images of the same scene. We assess homography accuracy by measuring corner correctness. We warp the four corners of a reference image to another using estimated homography and ground truth homography respectively and calculate the corner error between the warped points, as in [52, 59].

**Setup.** We evaluate on the widely used HPatches dataset [1]. HPatches comprises 52 sequences showing significant changes in illumination and 56 sequences displaying pronounced changes in viewpoint. Each sequence includes 1 reference image alongside 5 query images. We report the Area Under the cumulative Curve (AUC) of the corner error up to thresholds of 3, 5, and 10 pixels. OpenCV's RANSAC algorithm is used for robust estimation. We compare with two categories of methods: dense methods [10, 18, 28, 38, 45, 59, 63, 70, 72] and sparse methods [14, 44, 52, 52, 65, 75].

**Results.** Table 1 shows that PRISM outperforms other baselines under all error thresholds, which strongly proves the effectiveness of our method. We attribute the outstanding performance to the ability to capture multi-scale contexts provided by the SADPA. The iterative pruning paradigm also contributes to the accuracy by greatly reducing the mismatches.

### 4.2 Relative Pose Estimation

Relative pose estimation plays a fundamental role for various applications. We measure the relative pose error by the maximum angular error in rotation and translation. To determine the camera pose, we compute the essential matrix using predicted match points and apply both the RANSAC of OpenCV and LO-RANSAC of Poselib [26] for robust estimation.

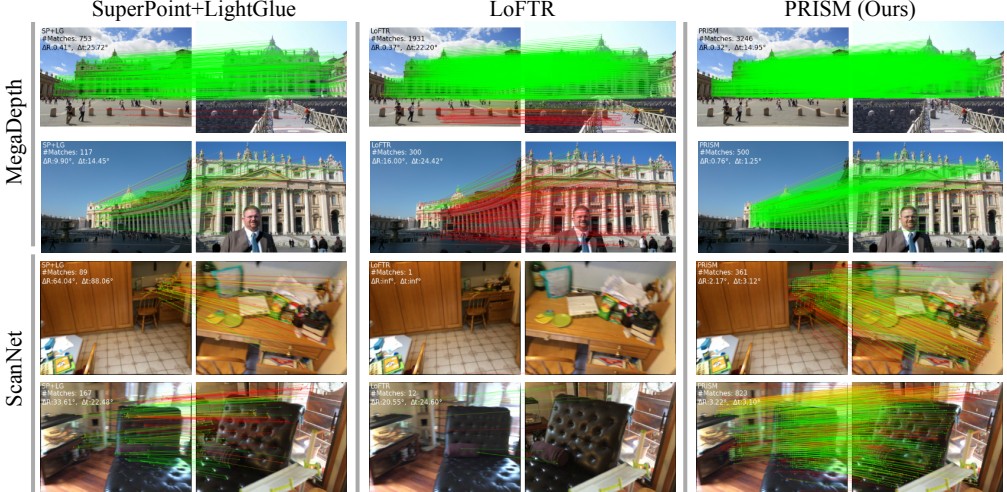

**Figure 4: Qualitative Results.** We compare PRISM with SP [13]+LG [32] and LoFTR [59] in ScanNet [11] and MegaDepth [29] dataset. As shown in the figure, PRISM can generate more dense matches and avoid most outliers in both indoor and outdoor scenes. The red color indicates epipolar error beyond $5 \times 10^4$ (in the normalized image coordinates). More visualizations are provided in the Appendix.

| | Methods | RANSAC AUC | | | Lo-RANSAC AUC | | |
|---|---|---|---|---|---|---|---|
| | | @5° | @10° | @20° | @5° | @10° | @20° |
| Sparse | SP [13]+SG [52] | 42.2 | 61.2 | 76.0 | 65.8 | 78.7 | 87.5 |
| | D2-Net [14]+NN | 19.6 | 36.3 | 54.6 | 33.4 | 47.3 | 60.4 |
| | R2D2 [44]+NN | 36.2 | 54.1 | 69.4 | 48.0 | 62.8 | 73.8 |
| | SP [13]+LG [32] | 49.4 | 67.2 | 80.1 | 66.3 | 79.0 | 87.9 |
| Dense | LoFTR [59] | 52.8 | 69.2 | 81.2 | 64.3 | 76.6 | 85.3 |
| | Quadtree [63] | 54.6 | 70.5 | 82.2 | 65.6 | 78.0 | 86.5 |
| | ASpanFormer [10] | 55.3 | 71.5 | 83.1 | 68.1 | 80.0 | 88.3 |
| | 3DG-STFM [38] | 52.6 | 68.5 | 80.0 | - | - | - |
| | AdaMatcher [22] | 52.4 | 69.7 | 82.1 | 64.1 | 76.8 | 85.6 |
| | ASTR [72] | 58.4 | 73.1 | 83.8 | - | - | - |
| | TopicFM [18] | 58.2 | 72.8 | 83.2 | 64.1 | 76.7 | 85.6 |
| | EfficientLoFTR [70] | 56.4 | 72.2 | 83.5 | 67.5 | 79.1 | 87.0 |
| | PRISM (ours) | **60.0** | **74.9** | **85.1** | **68.8** | **80.6** | **88.9** |

**Table 2: Relative pose estimation on MegaDepth dataset.**

| | Methods | RANSAC AUC | | |
|---|---|---|---|---|
| | | @5° | @10° | @20° |
| Sparse | D2-Net [14]+NN | 5.5 | 14.5 | 28.0 |
| | SP [13]+SG [52] | 16.2 | 33.8 | 51.8 |
| | SP [13]+OANet [73] | 11.8 | 26.9 | 43.9 |
| | SP [52]+LG [32] | 17.7 | 34.6 | 51.2 |
| Dense | DRC-Net [28] | 7.7 | 17.9 | 30.5 |
| | LoFTR [59] | 16.9 | 40.8 | 50.6 |
| | Quadtree [63] | 19.0 | 37.3 | 53.5 |
| | ASpanFormer [10] | 19.6 | 37.7 | 54.4 |
| | ASTR [72] | 19.4 | 37.6 | 54.4 |
| | TopicFM [18] | 17.3 | 34.5 | 50.9 |
| | Patch2Pix [75] | 9.6 | 20.2 | 32.6 |
| | EfficientLoFTR [70] | 19.2 | 37.0 | 53.6 |
| | PRISM (Ours) | **23.9** | **41.8** | **58.9** |

**Table 3: Relative pose estimation on ScanNet dataset.**

*Setup*. We evaluate our PRISM model on MegaDepth [29] and ScanNet [11] for relative pose estimation. MegaDepth is an extensive outdoor dataset with 1 million images across 196 scenes, reconstructed using COLMAP [54]. For testing, we use the same 1500 pairs as in [59], resizing images to a longer dimension of 1152. ScanNet is usually used for indoor pose estimation. It features monocular sequences with ground truth data and is challenging due to wide baselines and textureless regions. We follow the protocol of [59], resizing images to 640x480. To verify the PRISM's generalizability, the model is only trained on MegaDepth and tested on both datasets. Following [52, 59], we report the AUC of pose error at thresholds of 5°, 10°, and 20°. Note that we use the official codes, configurations and pre-trained weights to report the AUC under the Lo-RANSAC solver. We compare with dense methods [10, 18, 22, 28, 38, 59, 63, 70, 72, 75] and sparse methods [13, 14, 32, 44, 52, 73].

*Results*. Table 2 and Table 3 provide the AUC of pose error for MegaDepth and ScanNet, respectively. As we can see, our proposed PRISM achieves new state-of-the-art performance for all evaluation metrics. Thanks to the proposed Multi-scale Pruning Module, PRISM can avoid a large number of incorrect matches and perceive multi-scale information. Figure 4 qualitatively demonstrates our method's performance against others. For the ScanNet dataset, our method notably improves by 4.3% in AUC@5° and 4.5% in AUC@20° compared to the best model trained on MegaDepth, indicating the impressive generalization capability of our method.

### 4.3 Visual Localization

Visual Localization is essential in computer vision. The percentage of pose errors satisfying both angular and distance thresholds is reported, as in the Long-Term Visual Localization Benchmark [64].

*Setup*. We evaluate PRISM on the InLoc [61] dataset for indoor scenes and the Aachen Day-Night v1.1 [53, 74] dataset for outdoor scenes. The InLoc dataset consists of 9,972 geometrically registered

| | Methods | DUC1 | DUC2 | overall |
|---|---|---|---|---|
| | | (0.25m, 10°) / (0.5m, 10°) / (1m, 10°) | | |
| Sparse | SP [13]+SG [52] | 49.0/68.7/80.8 | 53.4/**77.1**/82.4 | 68.6 |
| | ClusterGNN [56] | 47.5/69.7/79.8 | 53.4/**77.1**/84.7 | 68.7 |
| | SP [13]+LG [32] | 49.0/68.2/79.3 | **55.0**/74.8/79.4 | 67.6 |
| Dense | LoFTR [59] | 47.5/72.2/84.8 | 54.2/74.8/**85.5** | 69.8 |
| | ASpanFormer [10] | 51.5/73.7/86.0 | 55.0/74.0/81.7 | 70.3 |
| | Patch2Pix [75] | 44.4/66.7/78.3 | 49.6/64.9/72.5 | 62.7 |
| | ASTR [72] | 53.0/73.7/87.4 | 52.7/76.3/84.0 | 71.2 |
| | TopicFM [18] | 52.0/74.7/87.4 | 53.4/74.8/83.2 | 70.9 |
| | CasMTR [7] | **53.5**/76.8/85.4 | 51.9/70.2/83.2 | 70.2 |
| | PRISM (Ours) | 53.0/**77.8/87.9** | 54.2/72.5/83.2 | **71.4** |

**Table 4: Indoor visual localization on InLoc dataset.**

RGBD indoor images and 329 query images with verified poses, posing challenges in textureless or repetitive environments. We use the two scenes named DUC1 and DUC2 for test as in [10, 59]. For outdoor localization, the Aachen dataset provides 6,697 daytime and 191 nighttime images, highlighting the difficulty of matching under significant illumination changes, especially at night. The metrics of the daytime and nighttime divisions are reported. We follow the guidelines provided by the Benchmark [64] to compute query poses. For both the indoor and outdoor datasets, candidate image pairs are identified using the pre-trained HLoc [50] system following [10, 32, 59]. Subsequently, camera poses are estimated utilizing our model, which was trained on the MegaDepth dataset. Dense methods [7, 10, 18, 22, 28, 40, 59, 70, 72, 75] and sparse methods [9, 13, 32, 52, 56] are compared.

**Results**. Table 4 presents the results for the InLoc dataset. PRISM is better than all baselines on DUC1 and on par with state-of-the-art methods on DUC2. Overall, we achieve the best performance on average. On Aachen V1.1 dataset, as shown in table 5, PRISM performs best on the day queries and the results on the night queries are slightly lower than that of LightGlue [32]. We argue that the Dense methods require quantification for the triangulation step of the localization pipeline, which harms the accuracy. Compared to dense methods, the performance of PRISM ranks among the top tier on night queries. In general, our method shows promising performances and strong generalization in visual localization tasks. These evaluations demonstrate our network's versatility across different task settings.

## 4.4 Understanding PRISM

**Ablation Study**. To fully understand the different design decisions in PRISM, we follow the same setting in Sec. 3.5 and conduct ablation experiments on MegaDepth dataset, as shown in Tab. 6. 1) Replacing the RoPE with the absolute positional encoding of LoFTR results in a degraded AUC. 2) Without the Patching Pruning module, there is a significant drop in pose estimation accuracy as expected. This demonstrates the efficacy of the proposed Patch Pruning module. 3) Using LoFTR's Linear Attention instead of SADPA leads to a noticeably declined result. We attribute this to the ability of SADPA to aggregate multi-scale information. 4) Replaceing SADPA with only single level attention at $\frac{1}{8}$ resolution will lead to degraded pose accuracy. It further validates the essentiality of the design of

| | Methods | Day | Night |
|---|---|---|---|
| | | (0.25m, 2°) / (0.5m, 5°) / (1m, 10°) | |
| Sparse | SP [13]+SG [52] | 88.2/95.5/98.7 | 86.7/92.9/**100** |
| | SGMNet [9] | 86.8/94.2/97.7 | 83.7/91.8/99.0 |
| | ClusterGNN [56] | 89.4/95.5/98.5 | 81.6/**93.9**/**100** |
| | SP [13]+LG [32] | 89.2/95.4/98.5 | **87.8**/**93.9**/**100** |
| Dense | LoFTR [59] | 88.7/95.6/99.0 | 78.5/90.6/99.0 |
| | ASpanFormer [10] | 89.4/95.6/99.0 | 77.5/91.6/99.5 |
| | AdaMatcher [22] | 89.2/95.9/99.2 | 79.1/92.1/99.5 |
| | PATS [40] | 89.6/95.8/**99.3** | 73.8/92.1/99.5 |
| | ASTR [72] | 89.9/95.6/99.2 | 76.4/92.1/99.5 |
| | TopicFM [18] | **90.2**/95.9/98.9 | 77.5/91.1/99.5 |
| | EfficientLoFTR [70] | 89.6/96.2/99.0 | 77.0/91.1/99.5 |
| | PRISM (Ours) | 89.4/**96.2/99.3** | 78.5/91.1/99.5 |

**Table 5: Outdoor visual localization on Aachen Day-Night v1.1 dataset.**

| Method | Pose estimation AUC | | |
|---|---|---|---|
| | @5° | @10° | @20° |
| 1)Replace RoPE to absolute positions | 57.3 | 73.0 | 83.9 |
| 2)Without Patch Pruning | 56.6 | 72.6 | 83.6 |
| 3)Replace SADPA with LoFTR's Attention | 55.3 | 70.9 | 82.9 |
| 4)Replace SADPA with single level design | 57.7 | 73.2 | 84.3 |
| 5)Without weighted Softmax | 58.5 | 73.4 | 84.1 |
| **Full** | **60.0** | **74.9** | **85.1** |

**Table 6: Ablation study on MegaDepth dataset.**

| Methods | resolution | | | |
|---|---|---|---|---|
| | 640 × 640 | 832 × 832 | 960 × 960 | 1152 × 1152 |
| LoFTR [59] | **89.6**/11.1 | **107.7**/17.3 | **145.1**/22.3 | 212.5/13.9 |
| AspanFormer [10] | 119.3/16.7 | 173.0/20.4 | 208.3/22.5 | 289.2/13.9 |
| PRISM(Ours) | 99.4/**5.4** | 119.7/**7.7** | 153.3/**9.9** | **209.1**/**13.5** |

**Table 7: Impact of test image resolution on the MegaDepth dataset [29].** We report both the runtime(ms)/VRAM(GiB).

SADPA. 5) The absence of the weights (estimated NMI) of Softmax will lead to a certain degree of performance degradation.

**Impact of test image resolutions**. We test the runtime and VRAM of PRISM under different image resolutions and compare with LoFTR [59] and AspanFormer [10], as shown in Table. 7. All results are based on one single A100 GPU. For the runtime, PRISM is slightly slower than LoFTR and largely outperforms AspanFormer. In terms of VRAM, PRISM can save about half of the VRAM usage compared to baselines in most cases.

## 5 CONCLUSION

In this paper, we propose PRogressive dependency maxImization for Scale-invariant image Matching (PRISM). PRISM gradually prunes irrelevant patch features during the feature interactions. Meanwhile, in order to better handle the scale discrepancy, we propose the Scale-Aware Dynamic Pruning Attention to aggregate information from different scales via a hierarchical design. Extensive experimental results on a wide range of benchmarks demonstrate the effectiveness and generalization of PRISM. With careful engineering optimizations, PRISM's time efficiency can be further enhanced.

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
