# OpenReview forum: "PRISM: PRogressive dependency maxImization for Scale-invariant image Matching"
_acmmm.org/ACMMM/2024/Conference — MM2024 Poster_

### Official Review · Reviewer_WMNd · 2024-05-07

**Rating:** 3
**Confidence:** 4

**Summary:**

This paper proposes a detector-free image matching method that prunes unnecessary image patches and models the scene of various scales within the same attention mechanism. This method uses the Coarse-to-fine pipeline, which is very popular in the field of image matching. Extensive experiments demonstrate its effectiveness on multiple datasets. From my point of view, the innovation of the paper is limited. It seems to only introduce mask and multi-scale information into the attention process. In the experiment section, many SOTA methods are not mentioned in this paper. Please refer to the limitation section for details. Overall, I think the quality of this paper is slightly below the threshold for acceptance.

**Strengths:**

1. This paper is well written and easy to understand.
2. Experiments are done on multiple public datasets, and the content is sufficient and detailed.
3. An innovative contribution lies in the proposal of the Patch Pruning module and the modification of the transformer module.

**Limitations:**

1. This paper lacks important baselines such as PMatch(CVPR 2023), DKM(CVPR 2023), RoMa(CVPR204). Comparisons should be made with them in the experimental section.
2. The core innovation seems to be simply introducing a mask through mutual information. The rest of the steps look similar to classical methods such as LoFTR.

**Suitability:**

3

---

### Official Review · Reviewer_pzeD · 2024-05-17

**Rating:** 5
**Confidence:** 3

**Summary:**

This paper aims to address two common issues:

1、Not all regions contribute effectively during the matching process: Introducing the Multi-scale Pruning Module (MPM) to adaptively prune irrelevant features by maximizing the dependency between the two feature sets.

2、Scale discrepancy：Proposing the Scale-Aware Dynamic Pruning Attention (SADPA) to aggregate information from different scales via a hierarchical design.

Extensive experiments across multiple datasets validate the effectiveness of the proposed approach.

**Strengths:**

1、The paper is well-structured and easy to follow.

2、The motivation of this paper is clear and reasonable. The proposed solution is simple yet effective.

3、The experimental parts are extensive and convincing, demonstrating high performance compared to many competitors. Furthermore, the ablation study validates the effectiveness of each component.

**Limitations:**

1、This method can prune irrelevant regions, which can reduce the matching area, thereby improving accuracy and reducing computational complexity, especially in scenarios with different scales or large viewpoint changes. However, how does the model perform when applied to simpler cases where there are not many regions that need pruning? Would the model's predicted masks filter out some beneficial regions? Since the model matching results are heavily dependent on the predicted pruning mask, more theoretical and experimental analysis of the quality of the predicted mask is required, including more visualization.

2、Although the proposed attention mechanism uses a multi-scale structure, there are no direct experiments conducted to verify the "scale-invariant" mentioned in the title. For instance, experiments could be performed to compare the performance of your method with other methods as the scale changes.

3、A discussion of limitations and failure cases is missing.

**Suitability:**

2

---

### Official Review · Reviewer_NWwY · 2024-05-22

**Rating:** 4
**Confidence:** 4

**Summary:**

This paper proposes a feature matching framework to tackle the scale variance between images and unnecessary feature computation in matching. This framework gradually prunes irrelevant patch features and tackles the scale discrepancy utilizing a novel attention structure.

**Strengths:**

+ The core idea is reasonable, removing non-covisible image features is helpful to both the accuracy and efficiency of matching.
+ The proposed attention structure is simple yet effective.
+ The manuscript is well-written and the experiments are convincing.

**Limitations:**

1. Scale variance in matching can also be addressed by scale estimation [1], overlap estimation [2] and area matching [3,4]. Also, as the core idea (avoiding computation in irrelevant image part) is highly similar to [3,4], a detailed comparison with them is important and should be added in this paper.
[1] Axel Barroso-Laguna, Yurun Tian, Krystian Mikolajczyk. ScaleNet: A Shallow Architecture for Scale Estimation; Proceedings of the IEEE/CVF Conference on Computer Vision and Pattern Recognition (CVPR), 2022, pp. 12808-12818
[2] Chen Y, Huang D, Xu S, et al. Guide Local Feature Matching by Overlap Estimation[C]//Proceedings of the AAAI Conference on Artificial Intelligence. 2022, 36(1): 365-373.
[3] Zhang Y, Zhao X, Qian D. Searching from Area to Point: A Hierarchical Framework for Semantic-Geometric Combined Feature Matching[J]. arXiv preprint arXiv:2305.00194, 2023.
[4] Zhang Y, Zhao X. MESA: Matching Everything by Segmenting Anything[J]. Proceedings of the IEEE/CVF Conference on Computer Vision and Pattern Recognition (CVPR), 2024.

2. Since the NMI is learned from the data, the equations about the concept are too dense in this version. Moreover, I wonder if a simple cosine similarity between patch features can achieve the same effect.

Minor:
3. Methods like LoFTR, ASpanFormer, TopicFM, etc, in table 4 are semi-dense methods, not dense methods.

I am willing to raise the score after the above concerns are addressed.

**Suitability:**

2

---

### Official Review · Reviewer_zdch · 2024-05-26

**Rating:** 3
**Confidence:** 3

**Summary:**

This paper uses Patch Pruning and multi-scale feature fusion to improve the performance of the LoFTR model in image matching tasks.

**Strengths:**

Experiments were conducted on multiple datasets, proving the effectiveness of the designed methods.

**Limitations:**

The paper does not describe the methods very clearly (I think Section 3 is not well-organized and somewhat confusing), nor does it clearly indicate the innovative aspects of the methods. In fact, I believe the authors mainly made improvements to the LoFTR[59] model, with the core improvements being Patch Pruning and multi-scale feature fusion. The authors need to clearly state that the base model framework used in this paper is LoFTR.

(1) For instance, the Self-Attention Layer and Cross-Attention Layer modules are already present in the original LoFTR, and are not newly proposed in this paper. The innovation of this paper lies in using Patch Pruning (i.e., Mask Map) and multi-scale feature fusion (three scales) in these two modules.

(2) Additionally, I want to know if previous image matching methods have used Patch Pruning. If so, how did they achieve this, and what are the advantages and disadvantages compared to Mutual Information (MI)? In fact, the authors did not clearly explain why MI was chosen for Patch Pruning.

**Suitability:**

2

---

### Meta-Review · Area_Chair_ewr3 · 2024-07-07

**Recommendation:** Accept (Poster)
**Confidence:** 4

**Metareview:**

This paper proposes a detector-free image matching method that eliminates unnecessary image parcels and models the scene at different scales as part of the same attention mechanism. Globally, the 4 reviewers indicate that the article is clearly written, well motivated; the contribution exists and is correctly evaluated through experiments. They also suggest to improve the state of the art to better position and evaluate this contribution, which could be done in the final version of the paper. But more important, it appears to be incremental with respect to another model (LoFTR), which may conduce to a publication as Poster, given the fact that the topic of image matching is at the core of many computer vision tasks.